# Modeling the consequences of age-linked rDNA hypermethylation with dCas9-directed DNA methylation in human cells

Yana Blokhina¤, Abigail Buchwalter[ID]*

Cardiovascular Research Institute and Department of Physiology, University of California, San Francisco, San Francisco, California, United States of America

¤ Current address: NewLimit, South San Francisco, CA, United States of America
* abigail.buchwalter@ucsf.edu

**Data Availability Statement:** All relevant data are within the manuscript and its supporting information files.

## Abstract

Ribosomal DNA (rDNA) genes encode the structural RNAs of the ribosome and are present in hundreds of copies in mammalian genomes. Age-linked DNA hypermethylation throughout the rDNA constitutes a robust "methylation clock" that accurately reports age, yet the consequences of hypermethylation on rDNA function are unknown. We confirmed that pervasive hypermethylation of rDNA occurs during mammalian aging and senescence while rDNA copy number remains stable. We found that DNA methylation is exclusively found on the promoters and gene bodies of inactive rDNA. To model the effects of age-linked methylation on rDNA function, we directed *de novo* DNA methylation to the rDNA promoter or gene body with a nuclease-dead Cas9 (dCas9)–DNA methyltransferase fusion enzyme in human cells. Hypermethylation at each target site had no detectable effect on rRNA transcription, nucleolar morphology, or cellular growth rate. Instead, human UBF and Pol I remain bound to rDNA promoters in the presence of increased DNA methylation. These data suggest that promoter methylation is not sufficient to impair transcription of the human rDNA and imply that the human rDNA transcription machinery may be resilient to age-linked rDNA hypermethylation.

## Introduction

Ribosomal DNA (rDNA) genes encode the structural RNAs of the ribosome and are present in hundreds of copies in mammalian genomes. Deregulation of the rDNA repeats is a conserved feature of aging that has been described in organisms including yeast, nematodes, flies, mice, and humans [1–4]. Diverse genetic and dietary manipulations that extend lifespan modulate rDNA activity [2, 5], suggesting that the major lifespan-regulating pathways exert their effects in large part by modulating ribosome biogenesis. The mammalian rDNA repeats accumulate DNA cytosine methylation (CpG methylation) throughout their regulatory and coding sequences during aging, and this pattern of methylation events can be read out as a highly robust "methylation clock" that correlates with biological age in humans, rodents, and canids [6, 7]. While methylation clocks have recently come to prominence as a metric to predict the

**Funding:** "We would like to acknowledge the following sources of funding: the UCSF Bakar Aging Research Institute (A.B.), NIH T32 in Molecular and Cellular Basis of Cardiovascular Disease (5T32HL007731, Y.B.), and the Glenn and American Federation for Aging Research Junior Faculty Award (A.B.). The funders had no role in study design, data collection and analysis, decision to publish, or preparation of the manuscript."

**Competing interests:** the authors have declared that no competing interests exist.

progression of aging, many of these clocks track methylation events at disparate sites scattered across the genome [8] that may share a similar chromatin state [9]. The rDNA methylation clock, in contrast, tracks methylation events at a specific locus with an essential function: supplying rRNAs for ribosome production. However, it remains unclear why the mammalian rDNA accumulates pervasive methylation with age or whether these methylation events have a consequence on rDNA function.

Here, we explore the relationships between rDNA methylation and transcription by re-mapping existing genomic data to the rDNA repeat, and we model age-linked rDNA hypermethylation in human cells by using dCas9 to direct DNA methylation to rDNA loci. While the deposition of methylation on mammalian rDNA promoters has been shown to inhibit transcription [10], the relationship between rDNA gene body methylation and activity has been unclear. We determine that active, RNA polymerase I (Pol I)-engaged rDNA repeats are hypomethylated throughout their regulatory and coding sequences, while silent repeats are densely methylated. While this correlation might suggest that age-linked hypermethylation would lead to silencing, we report the surprising finding that induction of rDNA methylation with a dCas9-methyltransferase fusion has no effect on rRNA transcription, nucleolar morphology, or cellular growth rate. Instead, we find that human UBF and Pol I are not displaced by rDNA promoter methylation but remain bound to methylated rDNA. These data suggest that the human rDNA transcription machinery may be resilient to age-linked rDNA hypermethylation.

## Results

### Age-linked hypermethylation of rDNA in mammalian aging

To visualize the density and magnitude of age-linked rDNA hypermethylation, we re-mapped a selection of reduced representation bisulfite sequencing (RRBS) and whole genome bisulfite sequencing (WGBS) datasets from young and aged mouse and human samples to the consensus rDNA repeat sequence. This approach provides a snapshot of the average methylation state of the hundreds of rDNA repeats present in mammalian genomes, which are nearly identical in sequence and organization [6, 11] (Fig 1A). First, we re-mapped a mouse blood RRBS dataset that was originally used to develop the rDNA methylation clock [6, 12] to the 45 kilobase-long consensus mouse rDNA repeat. We compared average rDNA CpG methylation in young mouse samples (n = 8, average age 3 months old) versus aged mouse samples (n = 9, average age 34 months old) and confirmed a general trend of increased CpG methylation along the entire rDNA coding sequence (Fig 1B) consistent with previous analysis of this dataset [6]. An RRBS dataset from human hematopoietic stem cells (HSCs) isolated from young (19–30 years old, n = 7) and aged (63–71 years old, n = 5) donors [13] similarly revealed extensive rDNA hypermethylation along the entire coding sequence when re-mapped to the 43 kilobase-long human rDNA repeat (Fig 1C). Finally, we re-mapped a WGBS dataset from IMR90 human lung fibroblasts undergoing replicative senescence (n = 3 each proliferative and senescent) [14], which indicated particularly pervasive hypermethylation along the human rDNA promoter and coding region, where 67% of CpGs exhibited at least a 5% increase in methylation in senescent cells, while the non-transcribed intergenic spacer lost methylation during senescence (Fig 1D). Overall, these analyses confirm that mammalian rDNA repeats undergo hypermethylation both during aging and replicative senescence, in contrast to the general trend toward hypomethylation that is observed elsewhere in the genomes of aging and senescent cells [6, 14].

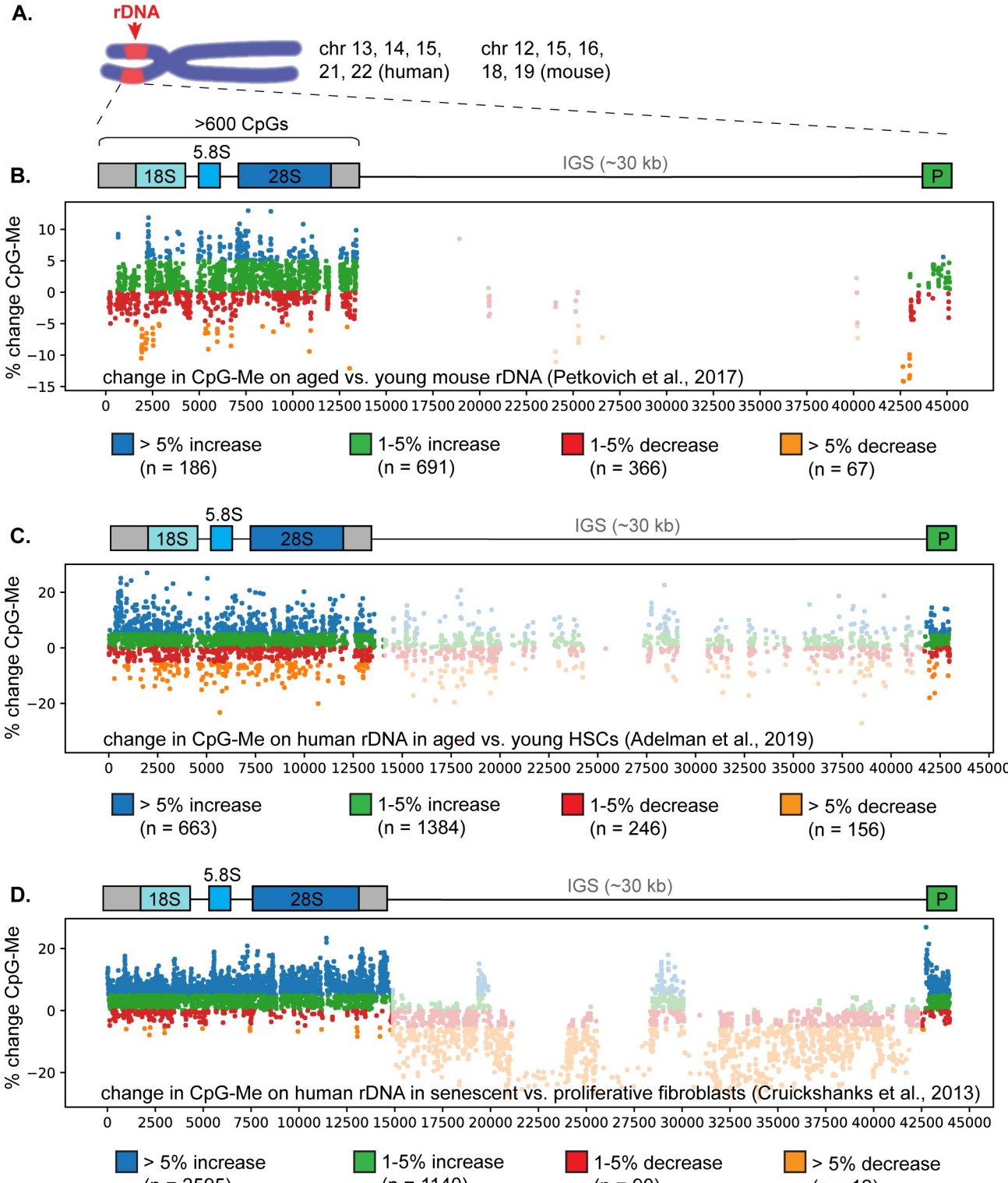

**Fig 1. Hypermethylation of the rDNA promoter and coding sequence in mammalian aging and senescence.** (A) Diagram of the location of rDNA arrays on mouse and human chromosomes. Each rDNA repeat is made up of a coding region that harbors > 600 CpG sites and encodes the 18S, 5.8S, and 28S rRNAs, followed by a long non-transcribed intergenic spacer (IGS). Note that the promoter (P) is positioned on the end of the rDNA repeat in rDNA reference sequences. Average % change in CpG methylation in (B) the rDNA of young mice (n = 8, average age 3 mo) versus aged mice (n = 9, average age 34 mo) measured by RRBS; (C) of human HSCs isolated from young (n = 7, 19–30 years old) or aged (n = 5, 63–71 years old) individuals

measured by RRBS; and (D) of proliferative (n = 3) versus senescent (n = 3) IMR90 human lung fibroblasts analyzed by WGBS. Note the hypomethylation of the non-transcribed intergenic spacer (IGS) in senescent fibroblasts, which is captured better by WGBS than by RRBS.

## Active human rDNA repeats are hypomethylated across the promoter and gene body

Age-linked rDNA hypermethylation occurs at hundreds of sites along the promoter and gene body of rDNA repeats [6] (Fig 1), but the consequences of this age-linked phenomenon remain largely a mystery. Under homeostatic conditions, a significant proportion of rDNA repeats are silenced by promoter methylation, while repeats lacking promoter methylation are transcribed by Pol I to produce rRNA [10]. The function of methylation within the rDNA gene body is comparatively poorly understood. In the case of genes transcribed by RNA Polymerase II (Pol II), CpG methylation on gene bodies is transcription-coupled and correlated in magnitude to transcription output [15, 16]. It is unclear if a similar relationship holds for the Pol I-transcribed rDNA. However, rDNA arrays exhibit mosaic methylation patterns, where completely unmethylated repeats and densely methylated repeats appear randomly interspersed [17, 18], which could suggest that completely unmethylated repeats are active, while methylated repeats are silenced.

To begin to understand the relationship between rDNA methylation state and Pol I transcription in humans, we re-mapped a recently published dataset of methylation-resolved ATAC sequencing (assay for transposable chromatin followed by bisulfite conversion and sequencing, or ATAC-Me) in human monocytes [19]. This method couples ATAC-seq, which identifies highly transcribed regions based on low nucleosome occupancy, to bisulfite sequencing, which identifies CpG methylation events. We found high ATAC-seq signal throughout the rDNA promoter and gene body (Fig 2C), consistent with previous observations that actively transcribed rDNA repeats have very few stably positioned nucleosomes [20]. CpG methylation occurs at a very low frequency on these ATAC-seq reads (~5%) throughout both the promoter and gene body (Fig 2B). In contrast, whole genome bisulfite sequencing (WGBS) of the same samples revealed dense CpG methylation across rDNA loci (Fig 2D and 2E). Taken together, these data imply that methylation of the rDNA promoter and gene body both occur exclusively on non-transcribed, silent repeats. However, this interpretation relies on inferring transcription activity indirectly from chromatin accessibility. To directly probe the methylation state of Pol I-engaged repeats in human cells, we immunoprecipitated Pol I-bound chromatin (S1 Fig), digested unmethylated bound DNA with the methylation-sensitive SmaI restriction enzyme, and detected the preserved methylated DNA by qPCR (a method referred to as "ChIP-chop" [21]). We evaluated the promoter, 5' ETS, 18S, 5.8S, and 28S rDNA regions using pairs of digestion-sensitive and digestion-insensitive primers to detect proportional CpG methylation at each locus (see Methods). At each of the analyzed sites within the promoter and within the gene body, we found that Pol I bound to hypomethylated rDNA (Fig 2F). These data demonstrate that actively transcribed (that is, Pol I-transcribed) rDNA repeats are hypomethylated at sites within their promoter, as has been previously established [10, 22], and further, are also hypomethylated along their transcribed sequence.

## rDNA copy number remains stable over lifespan in mammals

Our observation that the rDNA locus is either homogeneously methylated or unmethylated raises a possible explanation for age-linked hypermethylation of rDNA: changes in rDNA copy number (CN). That is, if methylated repeats are selectively retained while unmethylated repeats are selectively lost from aging genomes, the net effect would be an apparent increase in

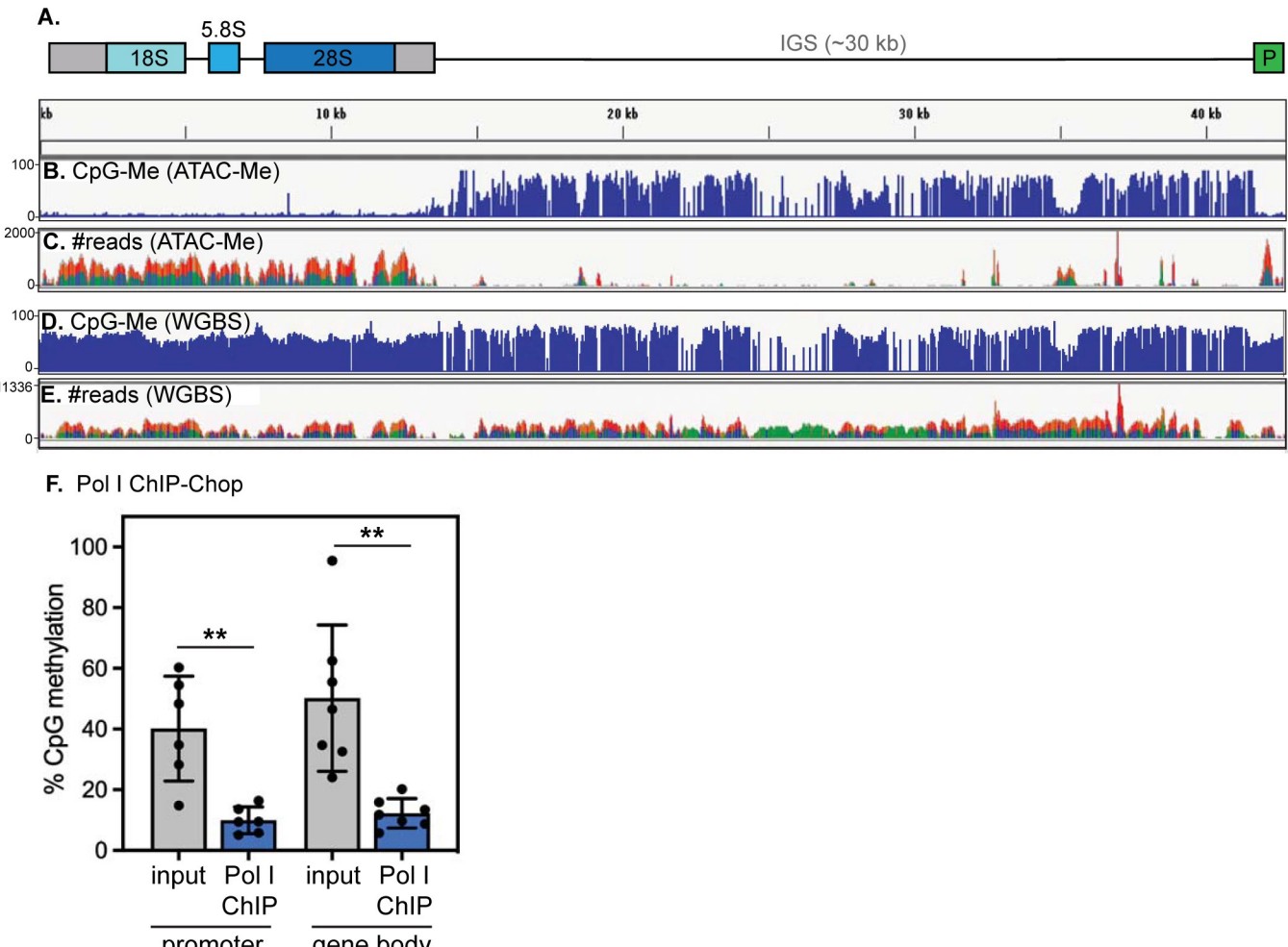

**Fig 2. Active human rDNA repeats are hypomethylated.** (A) Diagram of rDNA locus. (B) CpG methyl density on transposase accessible chromatin, detected by the ATAC-Me assay in human THP-1 monocytes. (C) ATAC-Me read density, correlated to chromatin accessibility, in the same dataset as in (B). (D) CpG methyl density on all rDNA repeats, detected by WGBS in human THP-1 monocytes. (E) WGBS read density on all rDNA repeats in the same dataset as in (D). Data re-analyzed from Barnett et al., *Mol Cell* 2020. (F) Analysis of %CpG methylation within actively transcribing, Pol I-bound rDNA repeats by Pol I ChIP-ChOP in HEK293T cells. When bound to Pol I, sites within the promoter (core promoter and 5'ETS) as well as within the gene body (18S, 5.8S, and 28S rDNA regions) are all hypomethylated (Pol I ChIP, blue) compared to the abundance of CpG methylation across all active and inactive rDNA loci (input, gray). N = 6. ** indicates that Pol I-bound rDNA is significantly less methylated than total rDNA at each locus analyzed (p < 0.005, unpaired t-test).

average methylation of rDNA repeats. Tandem repeats are vulnerable to CN changes during recombination events, as occurs during homology-directed DNA repair. Indeed, age-linked contraction of rDNA repeat arrays has been reported during replicative aging in yeast [1, 5] and organismal aging in flies [23]. However, evidence for rDNA CN change in mammals has been mixed [24, 25], and further complicating matters, rDNA CN varies dramatically across genetically diverse populations [26], which confounds analyses of rDNA CN in human populations. We surveyed rDNA CN in heart, quadriceps muscle, and liver tissues in adult (6–9 months of age) and aged (26 months of age) inbred C57Bl/6 mice (Fig 3A) and in the spleen of a separate cohort of C57Bl/6J animals at 6, 12, and 18 months of age (Fig 3B) by digital droplet PCR. These analyses clearly indicated that rDNA CN remains stable with age in mice. We infer from these findings that contraction of rDNA arrays does not underlie age-linked accumulation of rDNA CpG methylation in mammals. Instead, these findings increase the

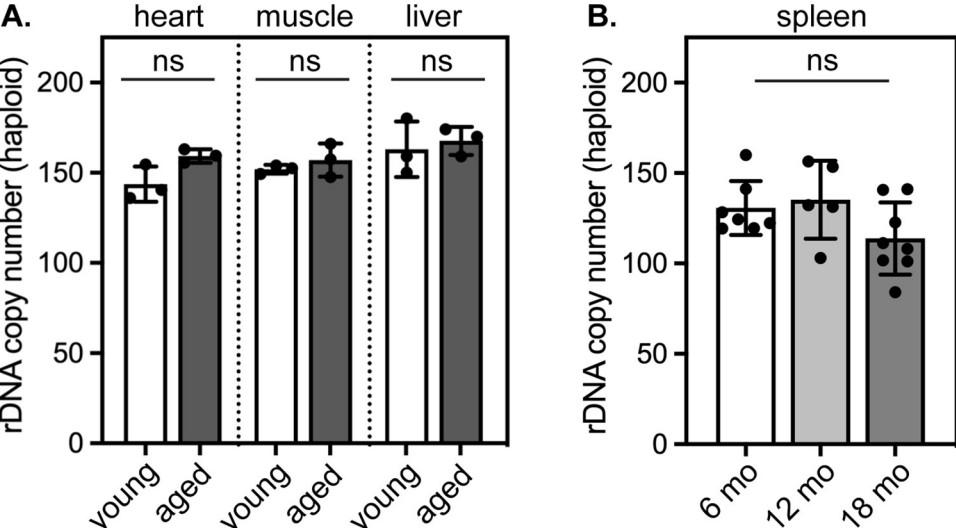

**Fig 3. rDNA copy number does not change over lifespan in mice.** (A-B) Quantification of rDNA copies per haploid genome by digital droplet PCR in the (A) heart, muscle, and liver of young (6–9 months) and old (26 months) C57Bl6/J mice (n = 3 mice per age group) and (B) in the spleen of 6 month old (n = 7), 12 month old (n = 5), and 18 month old C57Bl6/J mice (n = 8). ns in (A) indicates that all young-old comparisons are not significantly different (p > 0.05) by one-way ANOVA.

likelihood that age-linked hypermethylation arises as a consequence of increased methylation and/or impaired demethylation of a stable number of rDNA repeats.

## Inducing CpG methylation with a dCas9-methyltransferase fusion

We have determined that active rDNA repeats are completely hypomethylated while silent repeats are densely CpG methylated throughout both the promoter and gene body (Fig 2). As CpG methylation and rDNA silencing are correlated, one might predict that age-linked rDNA hypermethylation causes progressive silencing of rDNA repeats. Such an outcome could have major effects on the supply of ribosomes and on protein synthesis output in aging organisms. We chose to test this prediction in human cells by inducing CpG methylation at sites along the rDNA locus. To accomplish this goal, we used a DNMT3A-DNMT3L fusion protein that is directed to specific genomic loci via fusion to catalytically dead Cas9 (dCas9-3A3L) [27]. We adapted this genome editor by placing it under the control of a doxycycline-inducible promoter (Fig 4A) and stably integrated this expression cassette into HEK293T cell lines using the PiggyBac transposase [28]. We also stably expressed sgRNAs targeting sites within the rDNA locus as follows. We designed two sgRNAs targeting the upstream control element (UCE), a key regulatory region upstream of the rDNA core promoter: one site from -298 to -279 bp that we refer to as "P+G" and one site from -112 to -93 bp that we refer to as "P+A" (Fig 4B and S2 Fig, positions relative to transcription start site in GenBank U13369.1 human rDNA sequence). In addition, we designed a sgRNA targeting the 28S coding region (8349–8366 bp within the rDNA gene body) which we refer to as "28S+B", and a non-targeting control sgRNA (NTC) (Fig 4B). We validated inducible expression of dCas9-3A3L by Western blotting, and validated sgRNA-dependent targeting of the fusion protein to chromatin by ChIP-qPCR (S3 Fig). Next, we tested how effectively we could induce CpG methylation by inducing dCas9-3A3L expression for 72 hours, then performing methylation-sensitive qPCR at a site at the junction at the 3' end of the UCE (-54 bp) and at a site within the 28S coding region. We found that both the P+A guide and the P+G guide induced a 15–20% increase in methylation

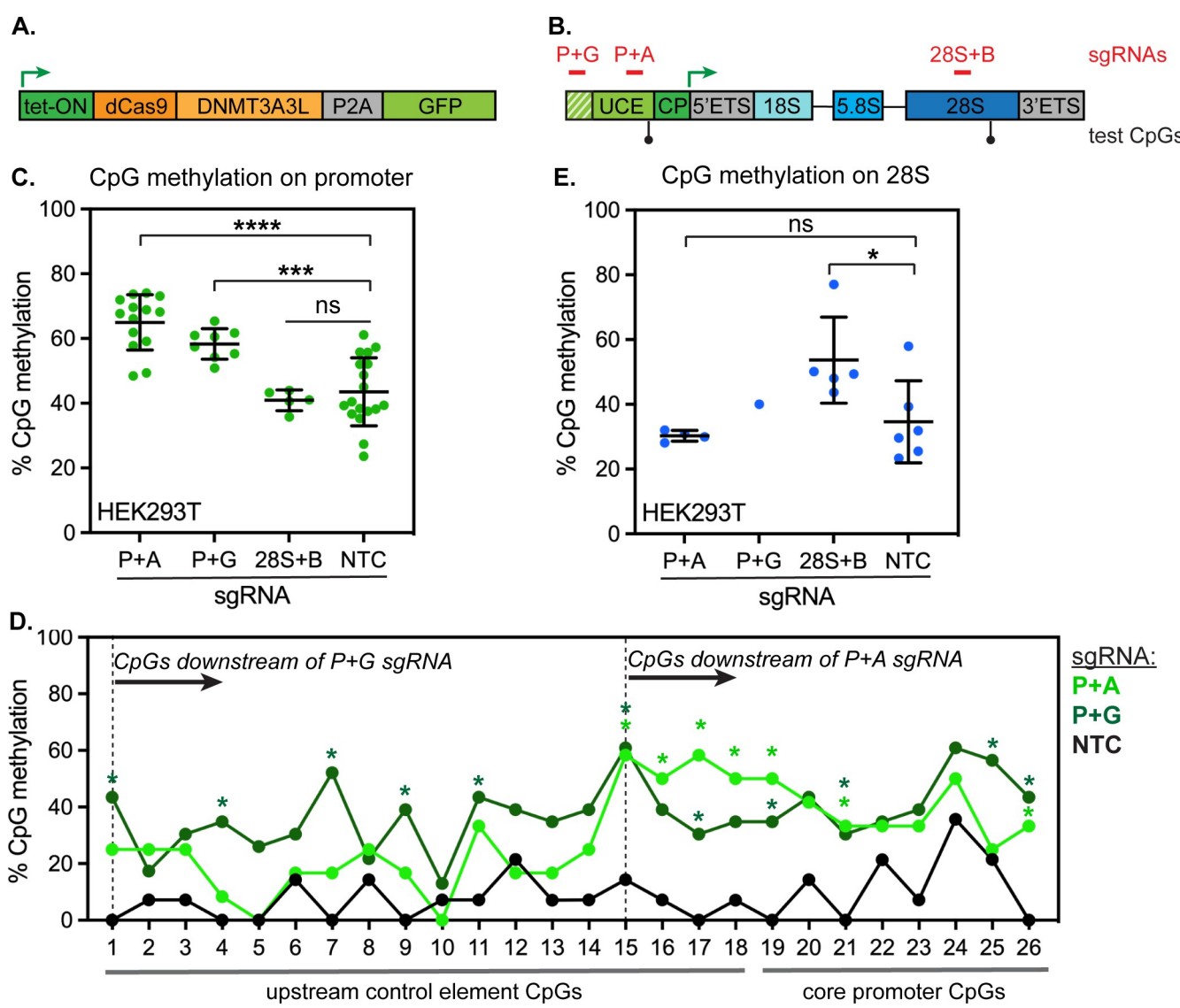

**Fig 4. Inducible methylation of the human rDNA locus with a dCas9-DNMT3A3L methyl editor.** (A) Diagram of tet-inducible dCas9-DNMT3A3L expression construct. (B) Diagram of human rDNA locus, showing positions of sgRNAs used within the promoter (P+G, P+A) and gene body (28S+B) of the locus. UCE, upstream control element; CP, core promoter; ETS, external transcribed spacer. (C) Detection of CpG methylation at the human rDNA promoter by methylation-sensitive qPCR (ChOP) of genomic DNA from HEK293T cells stably expressed for 72 hours. N = 14 for P+A; n = 8 for P+G; n = 5 for 28S+B; n = 18 for NTC. **** indicates p < 0.0001 and *** indicates p < 0.001 by unpaired t-test. (D) Detection of CpG methylation by bisulfite amplicon sequencing (BSAS) of 26 CpGs along the human rDNA UCE and core promoter in HEK293T cells induced to express dCas9-3A3L for 72 hours. N = 12 reads for P+A; n = 23 reads for P+G; and n = 14 reads for NTC. * indicates CpG positions with significantly increased methylation versus NTC (p < 0.05, Fisher's exact test). (E) Detection of CpG methylation on the human 28S rDNA by methylation-sensitive qPCR (ChOP) of genomic DNA from HEK293T cells induced to express dCas9-DNMT3A3L for 72 hours. N = 3 for P+A; n = 1 for P+G; n = 5 for 28S+B; n = 6 for NTC. * indicates p < 0.05 by unpaired t-test.

at the promoter CpG detection site (Fig 4C). Notably, the P+A guide is positioned ~50 base pairs upstream from the promoter CpG detection site, while the P+G site is ~250 base pairs farther upstream (Fig 4B); this indicates that dCas9-3A3L can modify sites several hundred base pairs away, as has been previously reported [27]. We profiled CpG methylation across the rDNA promoter by bisulfite conversion followed by amplicon sequencing (BSAS). This analysis indicated that the P+G guide significantly increased methylation of 11 CpGs downstream

from its recognition site throughout both the UCE and core promoter, while the P+A guide significantly increased methylation of 7 CpGs downstream of its recognition site at the 3' end of the UCE and within the core promoter (Fig 4D). However, we found no evidence that CpG methylation could spread to more distal sites, as the methylation state of the 28S gene body remained unchanged when methylation was induced at the promoter (Fig 4E). Conversely, we could induce a 15–20% increase in methylation within the 28S gene body using the 28S+B guide (Fig 4E), while methylation at the rDNA promoter remained unchanged (Fig 4C).

## Directed methylation of human rDNA does not impair transcription or ribosome biogenesis

Using the inducible dCas9-3A3L system, we were able to acutely increase rDNA methylation by 15–20% at each targeted spot within the rDNA locus (Fig 4)—an amount comparable to the increase of rDNA methylation observed during mammalian aging [6] (Fig 1). Surprisingly, however, we saw no evidence for impaired rRNA transcription after inducing methylation within either the promoter or the gene body, as the levels of both nascent pre-rRNA (Fig 5A) and mature 28S rRNA (Fig 5B) remained unchanged. Morphology of Pol I puncta within the dense fibrillar centers of nucleoli also remained normal, consistent with Pol I recruitment and transcription remaining unaffected (Fig 5C–5H). Finally, cell growth was unaffected by rDNA methylation, consistent with ribosome biogenesis remaining un-impaired (Fig 5I). These data indicate that direct CpG methylation of rDNA is not sufficient to induce rDNA silencing or to inhibit ribosome biogenesis. This outcome was unexpected, as inducing rDNA methylation in mouse cells has been shown to induce measurable silencing [29, 30], inducing methylation of individual CpGs in a human rDNA promoter reporter also causes measurable silencing [31], and dCas9-DNMT-controlled promoter CpG methylation can repress the transcription of Pol II-transcribed genes [32]. HEK293T cells are immortalized by transformation with the SV40 oncogene, which inactivates tumor suppressor genes including p53 and Rb and dramatically upregulates ribosome biogenesis [33]. To evaluate the consequences of induced rDNA methylation on a non-transformed, diploid cell line, we introduced inducible dCas9-3A3L and constitutively expressed sgRNAs into RPE1 cells. Again, we could induce a ~20% increase in rDNA promoter methylation after 72 hours of dCas9-3A3L expression (Fig 6A) but observed no corresponding changes to pre-rRNA abundance (Fig 6B), 28S rRNA abundance (Fig 6C), cellular growth rate (Fig 6D), or nucleolar morphology (Fig 6E and 6F). Altogether, these data indicate that increased CpG methylation is not sufficient to impair rRNA transcription in human cells.

## Directed methylation of the human rDNA promoter does not displace UBF

Transcription of rDNA is controlled by the essential transcription factor UBF, which binds to the rDNA core promoter and recruits Pol I. In mice, UBF is exclusively bound to rDNA repeats with unmethylated promoters, and CpG methylation of a single site within the core promoter is sufficient to displace UBF and inhibit transcription [10]. UBF exists in two alternatively spliced isoforms–UBF1 and UBF2. These isoforms heterodimerize with each other, but only UBF1 has been directly linked to control of rDNA transcription [34]. We evaluated UBF's binding to rDNA in HEK293T cells in the absence or presence of dCas9-3A3L-induced promoter CpG methylation induced by the P+A or P+G guides. These experiments were performed with an antibody that was recently shown to predominantly recognize UBF2 [35], but can immunoprecipitate both isoforms due to their heterodimerization [36]. We detected specific binding of UBF to the rDNA promoter but not to the 5S rDNA locus, which is transcribed by Pol III and is not bound by UBF (Fig 7A, 7C and S4 Fig). To our surprise, UBF remained

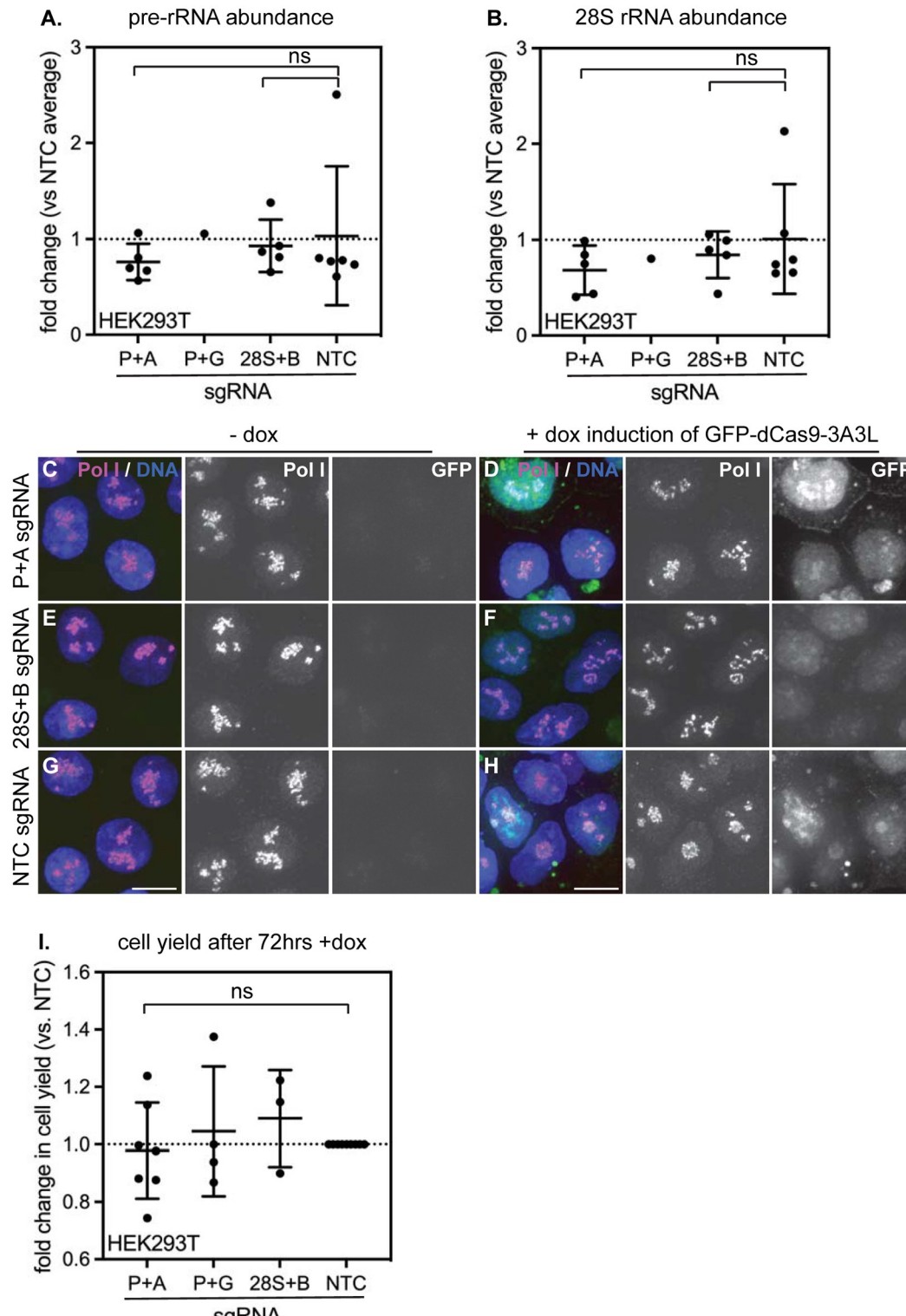

**Fig 5. Induced rDNA CpG methylation does not disrupt ribosome biogenesis, nucleolar morphology, or growth rate in HEK293T cells.** Quantification of 45S pre-rRNA abundance (A) and mature 28S rRNA abundance (B) by qPCR in HEK293T cells expressing the indicated sgRNAs and induced with doxycycline for 72 hrs. n = 5 for P+A; n = 1 for P+G; n = 5 for 28S+B; n = 6 for NTC. All ns (unpaired t-test). (C-H) Analysis of nucleolar morphology in HEK293T cells expressing the P+A sgRNA (C,D); 28S+B sgRNA (E,F); or NTC sgRNA (G,H) without doxycycline (C,E,G) or after 72 hrs

doxycycline induction of dCas9-DNMT3A3L (D,F,H). DNA stained with Hoechst dye (blue); Pol I stained with RPA194 antibody (red); methyl editor-expressing cells stained with GFP (green). Scale bar, 10 μm. (I) Quantification of cell numbers in cells expressing the indicated sgRNAs after 72 hrs of doxycycline induction. All ns (unpaired t-test).

bound to rDNA promoters in the presence of methylation (Fig 7A, 7C and S4 Fig). Instead, we found that UBF bound to more highly methylated rDNA after dCas9-3A3L induction in the presence of either the P+A or P+G guides (Fig 7B). These data clearly indicate that CpG methylation is not sufficient to displace UBF from the human rDNA promoter, and imply that

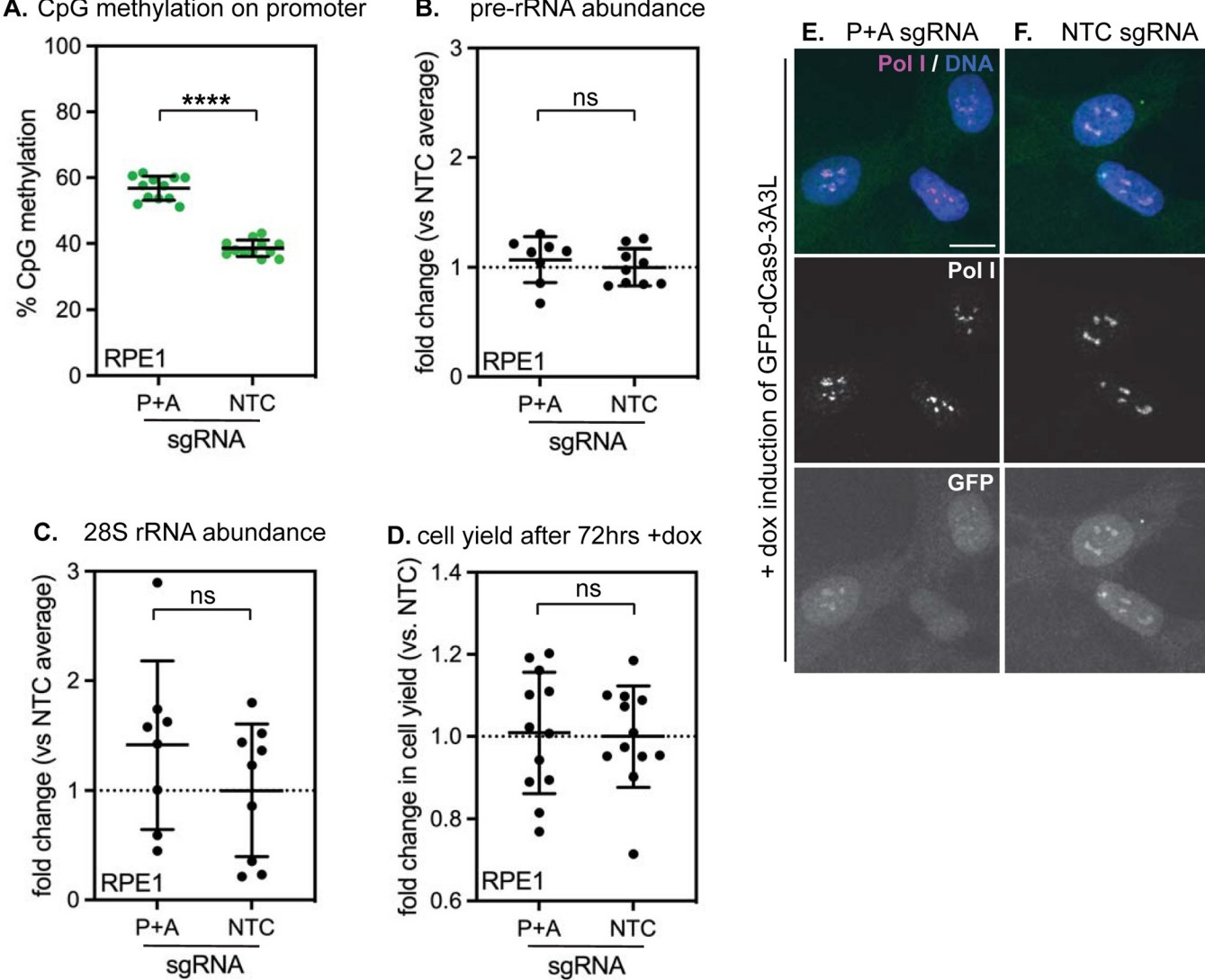

**Fig 6. Induced rDNA CpG methylation does not disrupt ribosome biogenesis, nucleolar morphology, or growth rate in RPE-1 cells.** (A) Detection of CpG methylation at the human rDNA promoter by methylation-sensitive qPCR (ChOP) of genomic DNA from RPE-1 cells stably expressing the indicated rDNA-targeting or non-targeting control (NTC) sgRNAs and induced to express dCas9-DNMT3A3L for 72 hours. N = 12 per condition. **** indicates p < 0.0001 (unpaired t-test). Quantification of 45S pre-rRNA abundance (B) and mature 28S rRNA abundance (C) by qPCR in RPE-1 cells expressing the indicated sgRNAs and induced with doxycycline for 72 hrs. n = 8 for P+A; n = 9 for NTC. All ns (unpaired t-test). (D) Quantification of cell numbers in cells expressing the indicated sgRNAs after 72 hrs of doxycycline induction. All ns (unpaired t-test). (E-F) Analysis of nucleolar morphology in RPE-1 cells expressing the P+A sgRNA (E) or NTC sgRNA (F) after 72 hrs doxycycline induction of dCas9-DNMT3A3L. DNA stained with Hoechst dye (blue); Pol I stained with RPA194 antibody (red); methyl editor-expressing cells stained with GFP (green). Scale bar, 10 μm.

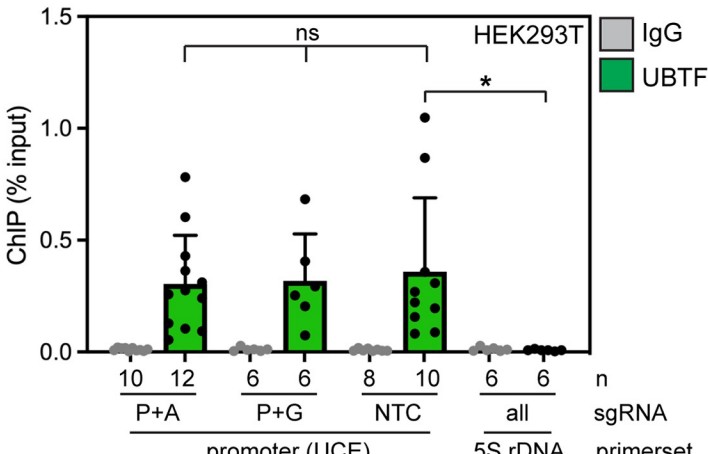

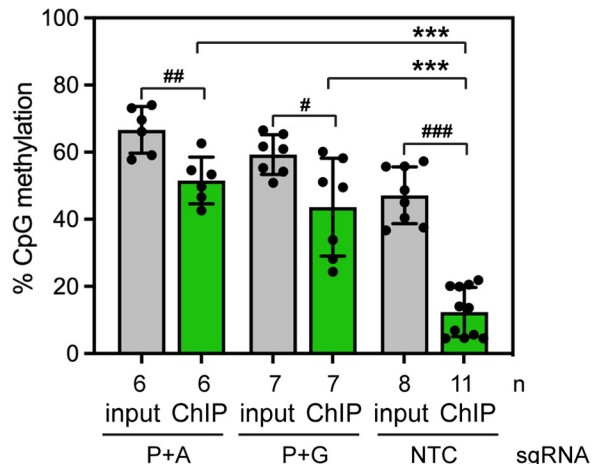

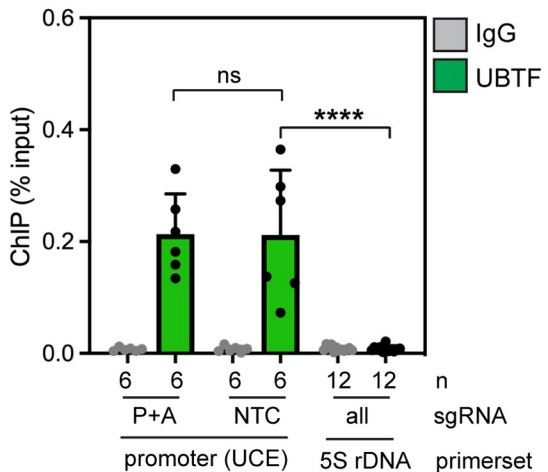

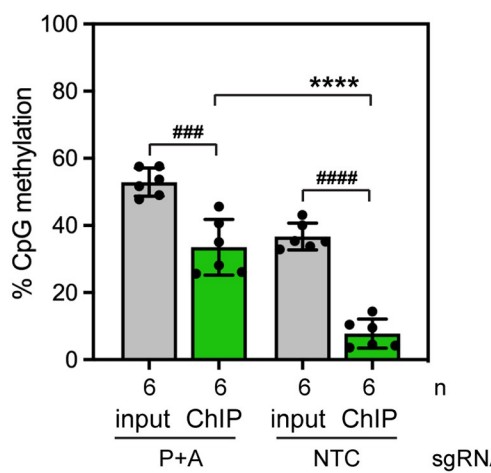

**Fig 7. Induced rDNA CpG methylation does not displace UBF from the human rDNA promoter.** (A) UBF ChIP-qPCR in HEK293T cells expressing the indicated sgRNAs and induced to express dCas9-DNMT3A3L for 72 hrs. UBF binding remains unchanged (ns, unpaired t-test) in the presence of rDNA promoter methylation; UBF binding at the promoter under all conditions is significantly enriched over background binding to the 5S rDNA locus (* indicates p < 0.05, unpaired t-test). (B) analysis of % CpG methylation within UBF-bound rDNA repeats by UBF ChIP-ChOP in HEK293T cells. #, ##, and ### indicate p < 0.05, 0.01, and 0.001 respectively for %CpGs on total DNA (gray) versus UBF-bound DNA (green). *** indicates p < 0.001 for % CpGs on UBF-bound DNA in cells expressing the NTC sgRNA versus the P+A or P+G sgRNA. (all analyses with unpaired t-test). (C) UBF ChIP-qPCR in RPE-1 cells expressing the indicated sgRNAs and induced to express dCas9-DNMT3A3L for 72 hrs. UBTF binding remains unchanged (ns, unpaired t-test) in the presence of rDNA promoter methylation; UBF binding at the promoter under all conditions is significantly enriched over background binding to the 5S rDNA locus (**** indicates p < 0.0001, unpaired t-test). (D) Analysis of %CpG methylation within UBF-bound rDNA repeats by UBF ChIP-ChOP in RPE-1 cells. ### and #### indicate p < 0.001 and p < 0.0001, respectively, for %CpGs on total DNA (gray) versus UBF-bound DNA (green). **** indicates p < 0.0001 for %CpGs on UBF-bound DNA in cells expressing the NTC sgRNA versus the P+A sgRNA.

unlike its mouse homolog, the human UBF transcription factor can bind to rDNA repeats in either methylation state. However, UBF-bound rDNA was still significantly less methylated than the total rDNA pool, suggesting that this transcription factor exhibits some preference for binding to unmethylated rDNA (Fig 7B). Pol I depends on UBF binding for recruitment to rDNA; consistently, we found that Pol I remained bound to rDNA repeats in HEK293T cells in the presence of CpG methylation (S4 Fig). We repeated these experiments in RPE1 cells expressing dCas9-3A3L and the P+A guide and similarly saw that UBF was not displaced

(Fig 7C) and instead bound to dramatically more methylated rDNA after induction of methylation (Fig 7D).

## Discussion

In this study, we sought to understand the functional consequences of age-linked rDNA hypermethylation in human cells. By evaluating relationships between methylation and transcription activity on human rDNA repeats, we determined that actively transcribed rDNA repeats are completely devoid of DNA methylation throughout both the promoter and coding sequence, while all rDNA repeats harbor dense CpG methylation in the IGS (Fig 2). Our data indicate that unmethylated repeats lack stably positioned nucleosomes and engage preferentially with Pol I (Fig 2). Consistent with our observations, recent visualization of patterns of CpG methylation along rDNA arrays by long-read sequencing indicated that rDNA repeats with completely demethylated coding sequences and completely methylated coding sequences are interspersed in rDNA arrays, while the IGS is uniformly methylated [18]. The recent completion of the human "Telomere to Telomere" genome assembly [37] will enable more in-depth exploration of the organization and regulation of rDNA arrays in the future.

The rDNA repeats may represent a weak point in the aging genome due to their repetitive nature, their extremely high transcriptional activity, and their poor protection by nucleosomes [20, 38] (Fig 2). We therefore considered the possibility that apparent rDNA hypermethylation could arise during aging by the selective loss of active, unmethylated rDNA repeats. In contrast to observations made in aging yeast [1, 5] and flies [23], we find that rDNA repeat CN remains completely stable through mammalian lifespan (Fig 3). Notably, a unifying feature of organisms that exhibit age-linked rDNA CN variation is the absence of DNA methylation. Human rDNA arrays, in contrast, are subject to DNA methylation and rapidly become unstable when DNA methyltransferase activity is disrupted [39], which suggests that DNA methylation may protect mammalian rDNA arrays from CN variation over lifespan.

We sought to model the effects of age-linked rDNA hypermethylation in human cells by directly inducing rDNA methylation and evaluating the consequences. While we were able to increase methylation by 15–20% at each targeted region within the human rDNA locus (Fig 4) —an amount comparable to the increase of rDNA methylation observed during mammalian aging–we were surprised to find no detectable consequence on rRNA transcription rates or on cell fitness (Figs 5, 6). In contrast, a similar level of methylation induction on the CpG island promoters of Pol II-transcribed targets via dCas9-3A3L causes measurable gene expression changes [32]. These contradictory outcomes could be explained by differences in the locus under study, differences in the mechanism of Pol I vs. Pol II transcription, or both. While dCas9-3A3L can repress transcription from a single-copy Pol II-transcribed gene [32], the rDNA repeats are multicopy loci; we cannot rule out the possibility that transcription is further accelerated on remaining hypomethylated rDNA repeats to compensate for dCas9-3A3L-deposited methylation and sustain rRNA synthesis. Nevertheless, our data indicate that human UBF and Pol I can tolerate significant levels of rDNA promoter methylation and remain bound to methylated rDNA repeats (Fig 7 and S4 Fig). However, we noted that UBF-bound rDNA does remain significantly less methylated than the total rDNA pool after methylation induction (Fig 7). This outcome is consistent with the previously documented preference of human UBF and Pol I for binding to less methylated promoters [40]. In contrast, the mouse Pol I machinery is sensitive to a single CpG methylation event in the UCE, which blocks UBF binding and prevents transcription [10]. The human rDNA promoter is significantly more CG-rich than the mouse rDNA promoter; it is possible that the human Pol I machinery has evolved to be more resilient to CpG methylation than the mouse Pol I machinery.

Transcription of a reporter gene under the control of the human rDNA promoter can be silenced by ~50% by methylation of individual promoter CpG sites [31]. In contrast, we were able to induce significant hypermethylation of 7–11 CpGs within the human rDNA repeat (Fig 4) but saw no effects on rDNA transcription (Fig 5). Importantly, plasmid reporter assays cannot assess the effects of CpG methylation on the full rDNA repeat in its endogenous genomic context. It is possible that CpG methylation is necessary, but not sufficient to displace the Pol I machinery from the human rDNA promoter, and that additional processes such as binding of methyl-CpG reader proteins, histone deacetylation, histone methylation, and/or nucleosome repositioning cooperatively discourage UBF and Pol I binding to rDNA [31, 41]. Altogether, these studies raise new questions about the regulation of transcription at human rDNA repeats and suggest that age-linked rDNA hypermethylation may have little effect on rDNA transcription output.

## Materials and methods

### Genomic data analysis

For re-mapping genomic data to the human ribosomal DNA, we used the following rDNA sequences: Genbank U13369.1 or the rDNA sequence extracted from BAC clone GL000220.1 [42]. GenBank BK000964.3 was used for re-mapping mouse genomic data to the mouse rDNA repeat. Accession numbers for published datasets re-analyzed in this study are as follows. RRBS from young and aged mouse blood [12]: GSE80672. RRBS from young and aged human HSCs [13]: GSE104408. WGBS from proliferative and senescent IMR90 lung fibroblasts [14]: GSE48580. ATAC-Me and WGBS from human THP-1 monocytes [19]: GSE130096. Quality control and adapter trimming were performed with TrimGalore (https://github.com/FelixKrueger/TrimGalore). Trimmed reads were aligned, processed to filter non-bisulfite-converted reads, and deduplicated with Bismark [43]. Methylated and unmethylated reads were each counted using the "bismark_methylation_extractor" script.

### rDNA copy number assessment

Genomic DNA was extracted from flash frozen mouse tissues obtained from the Nathan Shock Jackson Aging Center Pilot Project program using the PureLink Genomic DNA Mini Kit (ThermoFisher, Cat# K1820-02). DNA concentrations were measured on a Qubit fluorometer using the Qubit dsDNA HS Assay (Invitrogen) and 1 ng genomic DNA was used per 20 uL ddPCR reaction. Duplex ddPCR was performed using the droplet digital PCR supermix for probes (no dUTP) (Bio-Rad, Cat# 1863024) in the presence of the HaeIII restriction enzyme (VWR international, Cat# PAR6175); droplets were generated using the Bio-Rad QX200 droplet generator and quantified on the Bio-Rad QX200 Droplet Reader. Ribosomal DNA was amplified with primers rDNA2F and rDNA2R with probe rDNA2probe. rDNA Copy number was normalized to a single copy locus on chromosome 11 amplified with primers Chrom11F and Chrom11R with probe Chrom11probe (see S1 Table). Copy number was assessed using the Bio-Rad QuantaSoft software.

### Molecular cloning

To generate the plasmid XLone-dCas9-DNMT3A3L-P2A-EGFP, the dCas9-DNMT3A3L-P2A-EGFP region from plasmid pcDNA3.1-dCas9-Dnmt3a-Dnmt3l-P2A-eGFP (addgene# 128424) was amplified and inserted using the NEBuilder HiFi DNA Assembly Master Mix (NEB, Cat# M5520A) into the XLone-GFP backbone (addgene# 96930) digested with SpeI

(NEB, Cat# R0133S) and KpnI-HF (NEB, Cat# R3142S) to replace the GFP in the XLone plasmid with the dCas9-DNMT3A3L-P2A-EGFP insert.

The pKH011 sgRNA expression plasmid was the generous gift of Karissa Hansen and Elphege Nora. The blasticidin selection cassette was replaced with puromycin selection cassette. The puromycin sequence was amplified from addgene plasmid 104321, and inserted using the NEBuilder HiFi DNA Assembly Master Mix into the pKH011 plasmid digested with NcoI-HF (NEB, Cat# R3193S) and XbaI (NEB, Cat# R0145S) to remove the blasticidin cassette. CRISPR guide oligos (see Materials Used for sequence information) were annealed in T4 ligase buffer with addition of T4 Polynucleotide Kinase (NEB, Cat# M0201S) and inserted into the pKH011-puro backbone by simultaneous digestion of the backbone with BbsI-HF (NEB, Cat# R3539S) and ligation with T4 DNA ligase (NEB, Cat# M0202S) in T4 ligase buffer.

## Cell culture and transfection

HEK293T cells were maintained at 37˚C with 5% $CO_2$ in Dulbecco's Modified Eagle's Medium (DMEM) supplemented with 10% Fetal bovine serum (FBS) and 1% Penicillin-Streptomycin (Pen-Strep). Cells were transfected with pXLone-dCas9-DNMT3A3L-P2A-EGFP together with a plasmid encoding the piggyBAC transposase using the Lipofectamine 2000 transfection reagent (ThermoFisher, Cat# 11668019), and selected with 6 µg/mL blasticidin (RPI Research Products, Cat# 3513-03-9). Expression of pXLone-dCas9-DNMT3A3L-P2A-EGFP was induced with 2 µg/mL Doxycycline (VWR, Cat# AAJ67043-AD) and GFP-fluorescent cells were enriched by fluorescence-activated cell sorting. Individual cell clones were picked and transfected with pKH011-puro guide plasmids together with piggyBAC transposase and selected with 2 µg/mL Puromycin (Millipore Sigma, Cat# 540411). Cells were subsequently maintained with 6 µg/mL Blasticidin and 2 µg/mL Puromycin. Expression of pXLone-dCas9-DNMT3A3L-P2A-EGFP for methylation experiments was induced with 2 µg/mL Doxycycline.

RPE1 cells were maintained in DMEM/F-12 1:1 supplemented with 10% FBS, 1% Pen-Strep, and 0.01 mg/mL Hygromycin B (VWR, Cat# 45000–806). Cells were co-transfected with pXLone-dCas9-DNMT3A3L-P2A-EGFP and different pKH011-puro guide plasmids together with the piggyBAC transposase plasmid via electroporation with the Neon Transfection System (ThermoFisher). Transfected cells were selected and maintained with 6 µg/mL Blasticidin and 6 µg/mL Puromycin. Transfections were done in three independent biological replicates per pKH011-puro guide plasmid. Expression of pXLone-dCas9-DNMT3A3L-P2A-EGFP for methylation experiments was induced with 2 µg/mL Doxycycline.

## Chromatin immunoprecipitation (ChIP)

Cells were detached from culture plates with trypsin (0.25% in HBSS, GenClone Cat# 25–510) and pellets collected by centrifugation (300–500 rcf, 3–5 minutes). Pellets were washed in PBS and resuspended in fixation buffer (50 mM Hepes pH 8.0, 1 mM EDTA, 0.5 mM EGTA, 100 mM NaCl). Cells were fixed in 1% PFA for 5–10 minutes at room temperature, followed by quenching with 0.125M glycine and incubation on ice for 5 minutes. Fixed cells were pelleted (1200 rcf, 5 minutes, at 4˚C), washed in PBS, and flash frozen in liquid nitrogen. Fixed, frozen pellets were stored at -80˚C until ready to proceed.

Frozen pellets were resuspended in rinse buffer 1 (50 mM Hepes pH 8.0, 140 mM NaCl, 1 mM EDTA, 10% glycerol, 0.5% NP40, 0.25% Triton X-100) and incubated on ice for 10 minutes. Cells were pelleted (1200 rcf, 5 minutes, at 4˚C) and resuspended in rinse buffer 2 (10 mM Tris pH 8.0, 1 mM EDTA, 0.5 mM EGTA, 200 mM NaCl). Cells were pelleted and washed twice in shearing buffer (0.1% SDS, 1 mM EDTA pH 8.0, 10 mM Tris HCl pH 8.0) and subsequently resuspended in shearing buffer with freshly added protease inhibitors (cOmplete,

EDTA-free, Roche Cat# 05056489001). ~10–20 million cells per shearing tube were sheared by sonication using either the Bioruptor (Diagenode UCD-200; sheared on high, 10 cycles of 30 seconds on/ 30 seconds off, repeated 3 times) or the Covaris ultrasonicator (S220; PIP 140, DF 5%, CPB 200, 15 minutes) to fragment sizes of ~200-400bp.

Sheared chromatin was cleared by centrifugation (20000 rcf, 15 minutes, at 4˚C), supernatant was collected, and 10% input set aside and stored at -20˚C. ~3–7 million cells were used in each immunoprecipitation reaction in IP buffer (50 mM Hepes/KOH pH 7.5, 300 mM NaCl, 1 mM EDTA, 1% Triton X-100, 0.1% DOC, 0.1% SDS). Cells were immunoprecipitated with 2–4 μg of antibodies overnight at 4˚C. Antibodies used for ChIP were as follows: UBTF (F-9) mouse monoclonal antibody (Santa Cruz, sc-13125); RPA194 (C-1) (Pol I subunit) mouse monoclonal antibody (Santa Cruz, sc-48385); Cas9 mouse monoclonal antibody (Active Motif, Cat # 61757); and mouse pre-immune IgG (Sigma Aldrich Cat # 12–371). Samples were subsequently incubated with 20–40 μl Dynabeads M-280 sheep anti-mouse IgG (Thermo-Fisher, Cat# 11201D) for 2–4 hours at 4˚C. Beads were then washed twice in IP buffer, once in DOC buffer (10 mM Tris pH 8.0, 0.25 M LiCl, 0.5% NP40, 0.5% DOC, 1 mM EDTA), and once in TE buffer, for ~3 minutes per wash at room temperature. Chromatin was eluted from the beads twice for 20 minutes each in 150 μl Elution Buffer (1% SDS, 0.1 M NaHCO$_3$), and the eluates were combined.

The input material and eluted chromatin were subsequently treated with RNase, Proteinase K, and de-crosslinked overnight as follows. Input material volume was brought up to 100 μl in TE with a final concentration of SDS at 1%. Input and eluted chromatin were treated for 30 minutes at 37˚C with 0.2 μg/μl RNase A (ThermoFisher, Cat# EN0531). Then 6 μl of 0.5 M EDTA, 12 μl 1 M Tris (pH ~6.6) were added to eluted chromatin, and the eluted chromatin and input were treated for 2 hours at 55˚C with proteinase K (supplied with PureLink Genomic DNA Mini Kit, ThermoFisher, Cat# K1820-02). The input and eluted chromatin were subsequently de-crosslinked overnight at 65˚C. The samples were cleaned with DNA Clean & Concentrator-5 (Zymo Research, Cat# ZD4014) and eluted in ~10–30 μl nuclease free water. This material was subsequently used for enzymatic digestion for methylation assessment and quantitative PCR.

ChIP efficiency was quantified by the corrected % input method, where average Ct values of input and ChIP samples for each primerset used were adjusted for dilution factor as follows:

corrected Ct = Ct–log2(dilution factor)

dCt[ChIP] = corrCt[ChIP]–corrCt[input]

corrected %input = 100/(2^dCt[ChIP])

## Genomic DNA & RNA extraction from cells

Cells were treated with trypsin (0.25% in HBSS, GenClone Cat# 25–510) and cell pellets collected by centrifugation (300–500 rcf, 3–5 minutes). Genomic DNA was extracted from frozen or fresh pellets using the PureLink Genomic DNA Mini Kit (ThermoFisher, Cat# K1820-02) and eluted in nuclease free water. RNA was extracted from frozen or fresh pellets using the TRIzol reagent (ThermoFisher, Cat# 15596026) and eluted in nuclease free water.

## Enzymatic digest for methylation assessment of gDNA (Chop) and ChIP samples (ChIP-Chop)

Genomic DNA was sonicated using the Covaris (S220) prior to digestion with the following settings: PIP 140, DF 10%, CPB 200, 80s. ChIP samples and inputs were digested after de-crosslinking and cleaning as described under Chromatin Immunoprecipitation section above. Samples were digested in a 20 μl reaction with NEB Cutsmart buffer for 30 minutes—1 hour

**Table 1.**

| Sample Type | Sample amount | Cutsmart (µl) | Enzyme (µl) | Water | Water added at the end | Final Concentration |
|---|---|---|---|---|---|---|
| ChIP Input | 1 µl | 2 µl | 1 µl | Up to 20 µl | 20 µl | 1:40 |
| ChIP | 4 µl | 2 µl | 1 µl | Up to 20 µl | 20 µl | 1:10 |
| gDNA (sonicated) | ~10–100 ng | 2 µl | 1 µl | Up to 20 µl | variable | ~0.2–0.8 ng / µl |
| Control plasmid | 1 ng | 2 µl | 1 µl | Up to 20 µl | 180 µl | 0.005 ng / µl |

with SmaI (NEB, Cat# R0141) to assess rDNA promoter methylation or with HpaII (NEB, Cat# R0171) to assess 28S methylation. An unmethylated plasmid containing the rDNA promoter sequence and/or an unmethylated plasmid containing the HpaII cut site (pUC19, Addgene# #50005) were digested alongside the experiments to ensure efficient digestion. Digestion reactions were deactivated by incubation for 20 minutes at the deactivating temperature for each enzyme. Finally, water was added to each digest and the samples were analyzed by quantitative PCR. The samples were digested according to Table 1 below.

## Quantitative PCR (qPCR)

qPCR was performed using the SsoAdvanced Universal SYBR Green Supermix (Bio-Rad, Cat# 1725271) or the AzuraView GreenFast qPCR Blue Mix LR (Azura Genomics, Cat# AZ-2350). The reactions were run on a Bio-Rad real time PCR machine. All quantifications were normalized to standard curves made by 5-fold serial dilutions. For a list of primers used for methylation assessment and RT-qPCR see S1 Table.

## Methylation quantification

Methylation of genomic DNA and methylation of immunoprecipitated DNA were measured by Chop (methylation-sensitive PCR) and ChIP-chop [21], respectively. Briefly, each genomic DNA sample (for Chop) or ChIP sample (for ChIP-Chop) was amplified with two primer sets, one that amplified a fragment containing a target sequence for a methylation-sensitive enzyme (cut if not methylated), the other that amplified a fragment with no enzyme target sequence (never cut). % methylation was determined from the ratio of the levels of the former vs latter amplicon. See S1 Table for primer pairs used.

## Bisulfite amplicon sequencing

Genomic DNA was extracted from cells using the PureLink Genomic DNA Extraction Mini Kit (Invitrogen) according to the manufacturer's instructions. Purified genomic DNA was then bisulfite converted using the EpiJET Bisulfite Conversion Kit (Thermo Scientific) according to the manufacturer's instructions. The rDNA locus was amplified from the bisulfite-converted DNA using primers specific to bisulfite-converted sequences in the human rDNA locus; see S1 Table for primer pairs used. Amplified DNA was cloned into the pCR4-TOPO vector, then individual colonies were isolated, plasmid DNA was purified, and purified DNA was subjected to Sanger sequencing. Clones exhibiting incomplete cytosine conversion were discarded; completely converted clones were analyzed to determine the proportion of methylated to unmethylated CpGs.

## Immunocytochemistry / microscopy

Cells were seeded and grown on 8-well ibidi plates (Cat# 80826), then fixed within wells with 4% paraformaldehyde (PFA) in PBS for 3 minutes. The PFA was then removed and cells were

washed with PBS, incubated in IF buffer (0.1% Triton X-100, 0.02% SDS, 10 mg/mL bovine serum albumin, in PBS) for 20 minutes at room temperature (RT), and stained with RPA194 (C-1) (Pol I subunit) mouse monoclonal antibody (Santa Cruz, sc-48385) in IF buffer for 1 hour at RT. Cells were then washed with IF buffer, and stained with Alexa Fluor secondary antibodies (ThermoFisher) and Hoechst stain for 30 minutes in the dark at RT. Cells were washed and kept in PBS. Cells were imaged with a 63X 1.4NA oil-immersion objective on the Nikon CSU-X1 Spinning Disk microscope and images were analyzed using the ImageJ software.

## Supporting information

**S1 Fig. Pol I ChIP-qPCR at sites along the promoter and gene body of the human rDNA repeat in HEK293T cells. GAPDH, negative control.**
(EPS)

**S2 Fig. Relevant features of the human rDNA repeat (GenBank GL000220.1).**
(PDF)

**S3 Fig.** (A) Detection of dCas9-DNMT3A3L fusion in HEK293T cells induced with doxycy-cline. Predicted MW of fusion protein is ~225 kDa. (B) Cas9 ChIP-qPCR at rDNA promoter in untreated or doxycycline-induced HEK293T cells expressing the indicated sgRNAs. (C) Cas9 ChIP-qPCR at 28S rDNA in untreated or doxycycline-induced HEK293T cells expressing the indicated sgRNAs.
(EPS)

**S4 Fig.** (A) UBF ChIP-qPCR in HEK293T cells expressing the indicated sgRNAs and induced to express dCas9-DNMT3A3L for 72 hrs. A second independent promoter primerset used here versus data shown in Fig 6A. UBF binding remains unchanged (ns, unpaired t-test) in the presence of rDNA promoter methylation; UBF binding at the promoter under all conditions is significantly enriched over background binding to the 5S rDNA locus (* indicates $p < 0.05$, unpaired t-test). (B) UBF ChIP-qPCR in RPE-1 cells expressing the indicated sgRNAs and induced to express dCas9-DNMT3A3L for 72 hrs. A second independent promoter primerset used here versus data shown in Fig 6C. UBF binding remains unchanged (ns, unpaired t-test) in the presence of rDNA promoter methylation; UBF binding at the promoter under all conditions is significantly enriched over background binding to the 5S rDNA locus (**** indicates $p < 0.0001$, unpaired t-test). (C) Pol I ChIP-qPCR in HEK293T cells expressing the indicated sgRNAs and induced to express dCas9-DNMT3A3L for 72 hours. N = 2. (D) Analysis of % CpG methylation within Pol I-bound rDNA repeats by Pol I ChIP-ChOP in HEK293T cells. N = 2.
(EPS)

**S1 Table. List of primers used.**
(PDF)

**S1 Data. Datasets included in all graphs.**
(XLSX)

## Acknowledgments

We thank the Nathan Shock Jackson Aging Center Pilot Grant program for providing access to young and aged mouse tissues. We thank Karissa Hansen and Elphege Nora for sharing plasmid reagents; Daniele Canzio for shared use of equipment; and Devika Salim and Jennifer

Gerton for sharing digital droplet PCR probesets and protocols. Finally, we would like to thank all members of the Buchwalter, Nora, and Canzio labs for advice, support, and memes.

## Author Contributions

**Conceptualization:** Yana Blokhina, Abigail Buchwalter.

**Data curation:** Yana Blokhina, Abigail Buchwalter.

**Formal analysis:** Yana Blokhina, Abigail Buchwalter.

**Funding acquisition:** Abigail Buchwalter.

**Investigation:** Yana Blokhina, Abigail Buchwalter.

**Methodology:** Yana Blokhina, Abigail Buchwalter.

**Project administration:** Yana Blokhina, Abigail Buchwalter.

**Resources:** Abigail Buchwalter.

**Supervision:** Abigail Buchwalter.

**Visualization:** Abigail Buchwalter.

**Writing – original draft:** Yana Blokhina.

**Writing – review & editing:** Abigail Buchwalter.

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
