## [Decision Letter · Decision Letter 0]

23 Jul 2024

PONE-D-24-23946Modeling the consequences of age-linked rDNA hypermethylation with dCas9-directed DNA methylation in human cellsPLOS ONE

Dear Dr. Buchwalter,

Thank you for submitting your manuscript to PLOS ONE. After careful consideration, we feel that it has merit but does not fully meet PLOS ONE’s publication criteria as it currently stands. Therefore, we invite you to submit a revised version of the manuscript that addresses the points raised during the review process.

Your manuscript was reviewed by two of the referees who had reviewed your original manuscript submitted to the Review Commons. As you find in their comments, they still have several issues to be resolved before its publication. Please submit your revised manuscript by Sep 06 2024 11:59PM. If you will need more time than this to complete your revisions, please reply to this message or contact the journal office at plosone@plos.org. Please include the following items when submitting your revised manuscript:A rebuttal letter that responds to each point raised by the academic editor and reviewer(s). You should upload this letter as a separate file labeled 'Response to Reviewers'.A marked-up copy of your manuscript that highlights changes made to the original version. You should upload this as a separate file labeled 'Revised Manuscript with Track Changes'.An unmarked version of your revised paper without tracked changes. You should upload this as a separate file labeled 'Manuscript'.If applicable, we recommend that you deposit your laboratory protocols in protocols.io to enhance the reproducibility of your results. Protocols.io assigns your protocol its own identifier (DOI) so that it can be cited independently in the future. For instructions see: https://journals.plos.org/plosone/s/submission-guidelines#loc-laboratory-protocols. Additionally, PLOS ONE offers an option for publishing peer-reviewed Lab Protocol articles, which describe protocols hosted on protocols.io. Read more information on sharing protocols at https://plos.org/protocols?utm_medium=editorial-email&utm_source=authorletters&utm_campaign=protocols.

We look forward to receiving your revised manuscript.

Kind regards,

Hodaka Fujii, M.D., Ph.D.

Academic Editor

PLOS ONE

Journal Requirements:

2. Thank you for stating the following financial disclosure:"We would like to acknowledge the following sources of funding: the UCSF Bakar Aging Research Institute (A.B.), NIH T32 in Molecular and Cellular Basis of Cardiovascular Disease (5T32HL007731, Y.B.), and the Glenn and AFAR Junior Faculty Award (A.B.). ".  

3. Please expand the acronym “AFAR” (as indicated in your financial disclosure) so that it states the name of your funders in full.

4. Thank you for stating the following in the Acknowledgments Section of your manuscript: "We would like to acknowledge the following sources of funding: the UCSF Bakar Aging Research Institute (A.B.), NIH T32 in Molecular and Cellular Basis of Cardiovascular Disease (5T32HL007731, Y.B.), and the Glenn and AFAR Junior Faculty Award (A.B.). We thank the Nathan Shock Jackson Aging Center Pilot Grant program for providing access to young and aged mouse tissues. We thank Karissa Hansen and Elphege Nora for sharing plasmid reagents; Daniele Canzio for shared use of equipment; and Devika Salim and Jennifer Gerton for sharing digital droplet PCR probesets and protocols. Finally, we would like to thank all members of the Buchwalter, Nora, and Canzio labs for advice, support, and memes.

"

Please remove any funding-related text from the manuscript and let us know how you would like to update your Funding Statement. Currently, your Funding Statement reads as follows: "We would like to acknowledge the following sources of funding: the UCSF Bakar Aging Research Institute (A.B.), NIH T32 in Molecular and Cellular Basis of Cardiovascular Disease (5T32HL007731, Y.B.), and the Glenn and AFAR Junior Faculty Award (A.B.). "

Reviewers' comments:

Reviewer's Responses to Questions

**Comments to the Author**

1. Is the manuscript technically sound, and do the data support the conclusions?

Reviewer #1: Partly

Reviewer #2: Yes

2. Has the statistical analysis been performed appropriately and rigorously? 

Reviewer #1: Yes

Reviewer #2: Yes

3. Have the authors made all data underlying the findings in their manuscript fully available?

Reviewer #1: Yes

Reviewer #2: Yes

4. Is the manuscript presented in an intelligible fashion and written in standard English?

Reviewer #1: Yes

Reviewer #2: Yes

5. Review Comments to the Author

Reviewer #1: With reference to the reviews of this manuscript for Review Commons, I feel that within the scope of their study the authors have attempted, and in large part succeeded, adequately to respond to the reviewer’s comments.

However, I am of the opinion that the activity levels of the rDNA are far more important to aging than its methylation levels and feel the response to Reviewer 1 .1 comment simply attempts to dodge this key issue.

The response to comment of Reviewer 1 .5 concerning the UBTF antibody again simply attempts to sidestep the very important issue of specificity. I realize that logically the SC13125 antibody should, judging from its claimed derivation, detect both UBTF isoforms. Unfortunately, this is not the case experimentally, I refer to Theophanous et al. JBC 2023 Fig S1B. The counter argument that this antibody immunoprecipitates both UBTF forms is also fallacious since UBTF1 and 2 heterodimerize. But UBTF1 is the form that is active in rDNA expression and the limitation of using this antibody should be made clear. This said, the several ChIP experiments using the SC antibody may not be greatly in error due to UBTF heterodimerization.

Reviewer #2: The manuscript “Modeling the consequences of age-linked rDNA hypermethylation with dCas9-directed DNA methylation in human cells” describes an important finding about DNA methylation and its impact on rRNA transcription. The methods used are well suited and the models are also validated which is a strong point of the study. However, in addition to the previously answers to the concerns there are some validations that should be addressed.

Figure 1 and elsewhere – The published Genbank U13369.1 and other sequences have been used to study the rDNA locus. It was unclear whether the authors have tried to align the reads to the Genome assembly T2T-CHM13v2.0 to study transcription and DNA methylation of different loci? This should be made clear in the text.

Figure 7 – Altering the DNA methylation level at the rDNA did not change rRNA output, UBF-binding, nucleolar morphology or cell growth. The maintenance of transcription may be a result of opening more gene copies or by maintaining the transcription from the already open, active copies that are more methylated. A ChIP of RNA pol I or RRNA3 would be instrumental in assessing the level of polymerase binding at the promoter. Figure S4C shows a Pol I ChIP – is it to understand that no change in Pol I level occurred?

Figure 7 - Another factor that has been debated to have an impact on transcription is TTF-1, responsible for loop formation. Is the level of TTF-1 at the promoter changed?

Figure 7 - It would also be interesting to see the transcription from the upstream promoter, approximately 700-800 bp upstream of the TSS in human cells – does the higher DNA methylation at the promoter affect transcription upstream of the promoter? If any transcript level changes, that should be mentioned.

Supplementary Figure 1 – the variation in the ChIP experiment is high – is the highest binding an outlier?

6. PLOS authors have the option to publish the peer review history of their article (what does this mean?). If published, this will include your full peer review and any attached files.

Reviewer #1: No

Reviewer #2: **Yes: **Ann-Kristin Östlund Farrants

---

## [Author Response · Author response to Decision Letter 0]

12 Aug 2024

Response to Reviewers – Blokhina & Buchwalter – PONE-D-24-23946

We thank the reviewers and editorial staff for dedicating the time to re-review our manuscript. Our responses to each reviewer’s comment follow below.

R1.1: “I am of the opinion that the activity levels of the rDNA are far more important to aging than its methylation levels, and feel that the (original) response to Reviewer 1.1 comment simply attempts to dodge this key issue.”

We understand the reviewer’s point and agree that rDNA activity levels are clearly functionally important. We apologize that our original response to this point did not satisfactorily address the reviewer’s critique. The overarching goal of our study was to test how, if at all, age-linked rDNA CpG hypermethylation affects the function of human rDNA loci. It is exactly because of the point the reviewer makes – that activity levels of rDNA are what are most functionally important – that we set out to determine whether hypermethylation of human rDNA loci would have any measurable effect on rDNA activity. More generally, the relationship of age-linked “clock” CpG methylation sites to expression / function of CpG methylated loci is very unclear, and testing the potential relationship between age-linked rDNA methylation and function was the major goal of this study. Our finding that the human rDNA locus can tolerate hypermethylation to a similar extent as that observed during the aging process without any detectable changes to activity is relevant both to our understanding of human rDNA regulation and to our interpretation of DNA methylation clocks.

R1.2: “The comment… concerning the UBTF antibody again attempts to sidestep the very important issue of specificity.”

We thank the reviewer for providing an important and relevant reference (Theophanous et al., JBC 2023) which, as the reviewer describes, shows using RNAi that the Santa Cruz UBF antibody preferentially recognizes the UBF2 isoform over the UBF1 isoform. We have now cited this study in our discussion of the relevant experiments (Figure 7) in the Results section. Importantly, as the reviewer notes, UBF1 and UBF2 heterodimerize and thus immunoprecipitation of either isoform is likely to enrich some UBF1, the isoform that is capable of rDNA promoter TF activity. Indeed, we detect robust and specific UBF binding at the human rDNA promoter but not at other loci – such as the 5S rDNA locus - in our ChIP assays. Importantly, our interpretation that the human rDNA transcription machinery remains bound to rDNA in spite of CpG methylation is supported both by UBF ChIP (Fig. 7) and by Pol I ChIP (S4 Fig.). 

R2.1: “It was unclear whether the authors have tried to align the reads to the Genome assembly T2T-CHM13v2.0 to study transcription and DNA methylation of different loci?” 

We did not attempt to align sequencing reads to the Telomere to Telomere genome assembly. We agree that the T2T assembly provides exciting future opportunities for understanding the structure and regulation of rDNA repeat arrays, and we also cite recent work from the Kobayashi laboratory that used long-read sequencing approaches to resolve rDNA arrays. We now state in the Discussion that the T2T assembly will enable future dissections of rDNA repeats in their distinct genomic context.

R2.2: “A ChIP of RNA pol I or RRN3 would be instrumental in assessing the level of polymerase binding at the promoter. Figure S4C shows a Pol I ChIP – is it to understand that no change in Pol I level occurred?”

Yes, the reviewer is correct; we performed Pol I ChIP analyses in the absence or presence of dCas9-mediated hypermethylation and found that Pol I binding to the rDNA locus was unchanged. As we note in the Discussion, however, this average measurement of Pol I binding to rDNA promoters cannot rule out the “possibility that transcription is further accelerated on remaining hypomethylated rDNA repeats to…sustain rRNA synthesis. Nevertheless, our data indicate that human UBF and Pol I can tolerate significant levels of rDNA promoter methylation and remain bound to methylated rDNA repeats.”

R2.3: “Is level of TTF1 at the promoter changed?”

We thank the reviewer for this insightful question. We did not evaluate the binding of TTF1 to rDNA. Interestingly, work from the Moss laboratory indicates that TTF1 is the sole element of the rDNA transcriptional machinery that remains bound to the rDNA when UBF is acutely ablated (Hamdane et al., PLoS Genet 2014). We speculate, then, that TTF1 would remain bound to rDNA even if UBF and/or Pol I were displaced by DNA methylation. 

R2.4: “It would also be interesting to see the transcription from the upstream promoter…does the higher DNA methylation at the promoter affect transcription upstream of the promoter?”

The existence and function of a spacer promoter has been more clearly defined in the mouse rDNA repeat and has been little studied in the human rDNA repeat. Recent evidence indicates that the human Pol I transcription machinery binds a second location about 800 bp upstream of the core promoter in the human rDNA repeat (Mars et al G3 2018). This second locus does contain UBF1 binding sites and a core promoter motif, suggestive of its function as a spacer promoter. The guides that we used to direct CpG methylation recognize single unique sites in the core rDNA promoter that are not homologous to this putative spacer promoter, and we did not analyze methylation at the spacer promoter. 

To address the reviewer’s question of whether induced methylation at the core promoter could affect upstream transcription from the spacer promoter, we designed two primersets to amplicons just downstream of the putative transcription start site of the human spacer promoter described by Mars et al. and diagrammed in Figure 5 of that study. These primersets are:

Set 1: ~80 bp product immediately downstream of putative spacer promoter TSS

Fwd: TGG GGG AGA GGC TGT CG

Rev: CTA CGG CGC GCT GGT C

Set 2: ~100 bp product ~100 bp downstream of putative spacer promoter TSS

Fwd: TAG CTC CCG AGG CCC GA

Rev: GGC CAG GAT GAG CGG AC

We used these two primersets to evaluate rRNA abundance in HEK293T cells after induction of dCas9-DNMT and a non-targeting control (NTC) sgRNA vs. induction of dCas9-DNMT and our “P+A” core promoter-targeting sgRNA. (See S2 Fig for diagram of guide position.) Both of these primersets did amplify RNA, but did not detect any significant difference in RNA abundance in cells treated with the promoter-targeting guide compared to those treated with the non-targeting guide. (For reference, this guide induced >20% increase in promoter CpG methylation in HEK293Ts; see Fig. 4C). Overall, this outcome is consistent with the lack of effect on rRNA abundance that we report in Figure 5.

We thank the reviewer for raising this question and share the result of this experiment here; however, because so little is known about transcripts originating from the putative human spacer promoter, we feel that this analysis is too preliminary to be included in the manuscript.

R2.5: “In Supplementary Figure 1 - the variation in the ChIP experiment is high - is the highest binding an outlier?”

The data shown in S1 Fig are only intended to demonstrate the ability of primersets used to detect Pol I binding along the rDNA; while one dataset may be an outlier this does not affect our assessment that these primersets can effectively detect Pol I binding above IgG background along the rDNA repeat. These same primersets are used in Figure 2 in independent ChIP-Chop experiments.

---

## [Decision Letter · Decision Letter 1]

2 Sep 2024

PONE-D-24-23946R1Modeling the consequences of age-linked rDNA hypermethylation with dCas9-directed DNA methylation in human cellsPLOS ONE

Dear Dr. Buchwalter,

Thank you for submitting your manuscript to PLOS ONE. After careful consideration, we feel that it has merit but does not fully meet PLOS ONE’s publication criteria as it currently stands. Therefore, we invite you to submit a revised version of the manuscript that addresses the points raised during the review process.

Your revised manuscript was reviewed by two of the referees who had reviewed the original manuscript. They support its publication, but one of them pointed out a minor error in Figure 1. Please correct it, and I will make a final "accept" decision upon receipt of the revised manuscript.

We look forward to receiving your revised manuscript.

Kind regards,

Hodaka Fujii, M.D., Ph.D.

Academic Editor

PLOS ONE

Journal Requirements:

Reviewers' comments:

Reviewer's Responses to Questions

**Comments to the Author**

1. If the authors have adequately addressed your comments raised in a previous round of review and you feel that this manuscript is now acceptable for publication, you may indicate that here to bypass the “Comments to the Author” section, enter your conflict of interest statement in the “Confidential to Editor” section, and submit your "Accept" recommendation.

Reviewer #1: All comments have been addressed

Reviewer #2: All comments have been addressed

2. Is the manuscript technically sound, and do the data support the conclusions?

Reviewer #1: Yes

Reviewer #2: Partly

3. Has the statistical analysis been performed appropriately and rigorously? 

Reviewer #1: Yes

Reviewer #2: Yes

4. Have the authors made all data underlying the findings in their manuscript fully available?

Reviewer #1: Yes

Reviewer #2: Yes

5. Is the manuscript presented in an intelligible fashion and written in standard English?

Reviewer #1: Yes

Reviewer #2: Yes

6. Review Comments to the Author

Reviewer #1: (No Response)

Reviewer #2: The authors have satisfactory answered my concerns. However, i saw a small error in Figure 1. While the mouse rRNA gene repeat is 45 kb, the human one is 43 kb. Figure 2 this point seems to be corrected.

7. PLOS authors have the option to publish the peer review history of their article (what does this mean?). If published, this will include your full peer review and any attached files.

Reviewer #1: No

Reviewer #2: No

---

## [Author Response · Author response to Decision Letter 1]

3 Sep 2024

Final revision 9-3-24: There is some confusion on one remaining point from the reviewer about the size of the rDNA repeat (45 kb in mouse, 43 kb in human). In Figure 1, we re-map published mouse and human datasets to the mouse and human rDNA repeats, respectively. Figure 1B shows mouse rDNA repeat data, which is why there are sequencing reads mapped up until the 45 kb marker on the graph shown. Figures 1C-D show human rDNA repeat data, and consistently sequencing reads end just after the 42500 marker on the graphs shown. 

To make this point as clear as possible, we here submit a revised manuscript that clearly states in the corresponding Results section that the Fig. 1B shows mapping of sequencing data to the 45 kb mouse rDNA repeat while all subsequent genomic data are mapped to the human 43 kb rDNa repeat.

---

## [Editor Report · Decision Letter 2]

5 Sep 2024

Modeling the consequences of age-linked rDNA hypermethylation with dCas9-directed DNA methylation in human cells

PONE-D-24-23946R2

Dear Dr. Buchwalter,

We’re pleased to inform you that your manuscript has been judged scientifically suitable for publication and will be formally accepted for publication once it meets all outstanding technical requirements.

Kind regards,

Hodaka Fujii, M.D., Ph.D.

Academic Editor

PLOS ONE
---

## [Editor Report · Acceptance letter]

12 Sep 2024

PONE-D-24-23946R2 

PLOS ONE

Dear Dr. Buchwalter, 

I'm pleased to inform you that your manuscript has been deemed suitable for publication in PLOS ONE. Congratulations! Your manuscript is now being handed over to our production team.

Kind regards, 

on behalf of

Dr. Hodaka Fujii 

Academic Editor

PLOS ONE